# Determination of Vitamin B2 Content in Black, Green, Sage, and Rosemary Tea Infusions by Capillary Electrophoresis with Laser-Induced Fluorescence Detection

**Filiz Tezcan * and F. Bedia Erim**

Department of Chemistry, Istanbul Technical University, Maslak, Istanbul 34469, Turkey; erim@itu.edu.tr
* Correspondence: filiz.tezcan@acibadem.edu.tr; Tel.: +90-532-568-3669

**Abstract:** Vitamin B2, also known as riboflavin (RF) is an essential micronutrient for human health and must be obtained from dietary sources. Plants biosynthesize riboflavin and are important dietary sources of vitamin B2 for humans. Our present study reports sensitive detection of vitamin B2 in widely consumed tea infusions, namely black, green, sage and rosemary tea infusions, by a capillary electrophoresis method combined with laser induced fluorescence detection. Moreover, the correlation between the vitamin B2 content of tea plants with their total phenolics (TPs) and antioxidant capacity are evaluated in this study. Whereas green teas have the highest TPs and antioxidant capacity, the highest RF content is in sage infusions. The RF content ranged between 0.34 and 10.36 µg/g for all tea samples studied. Comparing the RF content of tea samples found in this study to the RF content of known RF sources, tea infusions are proposed as important dietary sources of vitamin B2.

**Keywords:** tea; Salvia officinalis; *Rosmarinus officinalis*; total phenolic; antioxidant

## 1. Introduction

Tea from the leaves of the *Camellia sinensis* plant is the most widely consumed drink in the world, after water. There are three kinds of tea products produced from the same plant: Black (fermented), green (not fermented) and oolong (partially fermented) teas. The tea leaves are dried, crushed and left for fermentation. Fermentation is a natural process and oxidation during fermentation is caused by a natural enzyme in the tea leaves themselves. While green tea is not fermented, black tea is subjected to full fermentation and the color of the tea leaves change to black. Tea is a dietary source of many bioactive compounds. Both black and green tea contain bioactive phenolic compounds mainly gallic, trans-cinnamic, caffeic, ferulic, p-coumaric acids, catechin, epicatechin, quercetrin, and caffeine [1,2]. However their content differs with different processing levels of tea samples. Several chemo metric studies have been developed for tea identification based on the bioactive ingredients and antioxidant activities of tea samples [1,2]. The health benefit effects of tea on cancer and cardiovascular diseases, to obesity and diabetes were extensively reviewed by Khan and Mukhtar [3]. Afzal et al. reviewed the potential therapeutic role of epigallocatechin gallate, which is the main constituent of green tea [4]. It was shown that this compound is associated with antitumor, anti-Alzheimer, and anti-aging properties. Heber et al. reported that all three tea polyphenol extracts induced weight loss and had anti-inflammatory and angiogenic effects [5].

In 2010, world tea production reached over 4.52 million tons [6]. Turkey is the fifth largest producer of tea in the world after China, India, Kenya and Sri Lanka [7]. Furthermore, Turkey has one of the highest per capita black tea consumption rates in the world. The attention given to green tea

in Turkey has been increasing due to its effect against obesity. Sage (*Salvia officials*) tea is one of the most popular herbal teas in Turkey. Like black and green teas, sage and rosemary contain many health beneficial bioactive compounds [8,9].

Riboflavin (RF), or its commonly known name vitamin B2, is a water-soluble vitamin and essential for human health. RF must be obtained from foods since it cannot be synthesized or stored in the body. Vitamin B2 deficiency affects many organs and tissues [10]. Milk, dairy products, meat, fish, dark-green vegetables, and some beverages like beer and wine are important sources of this vitamin. The analysis of RF in food samples is difficult because of the complex matrix of food and very low content of RF in foodstuffs. Recently, capillary electrophoresis (CE) has received great attention in food analysis due to its easy method, development availability, low sample consumption, fast analysis times, and inexpensive separation columns [11]. Lately, a combination of laser induced fluorescence (LIF) detectors with capillary electrophoresis has provided a remarkable improvement in detection limits. Since RF has a native fluorescence, the RF content of various foods have been studied by capillary electrophoretic methods using LIF detection [12–17].

Although a significant number of studies have been reported on tea, sage, and rosemary phenolics, almost no information exists concerning the vitamin content of these plants. Hu and coworkers reported vitamin B2 in two green tea samples [14]. To our knowledge, there is no study on the content of vitamin B2 in sage and rosemary. The aim of this study is to contribute to the information on the nutritional value of widely consumed tea infusions, by determining the vitamin B2 content of tea plants using the CE-LIF method. Moreover, the correlation between the vitamin B2 content of tea plants with their total phenolics and antioxidant capacity are evaluated in this study.

## 2. Materials and Methods

### 2.1. Materials

Riboflavin, Folin-iocalteu reagent, gallic acid, 2,4,6-tripyridyl-s-triazine and $FeCl_3 \cdot 6H_2O$ were from Sigma Chemical Co. (Steinheim, Germany). Di-sodium hydrogen phosphate dehydrates, sodium carbonate anhydrous, sodium acetate trihydrate, and $FeSO_4 \cdot 7H_2O$ were from Merck (Darmstadt, Germany). All solutions were prepared with water purified by an ElgaPurelab Option-7–15 model system (Elga, UK).

Four Black (B1–B4) and two green (G1 and G2) tea-bag samples were obtained from local markets inIstanbul as known commercial brands. The B1 sample consisted of teas from the East Black Sea region of Turkey. The others were tea blends. According to the labels, B2 was a blend of Kenyan and Sri Lankan teas. B3 was a blend of Turkish, Kenyan, and Indonesian teas. B4 was a blend of Turkish, Sri Lankan, Kenyan, and Indian teas. Sage and Rosemary samples were purchased from Istanbul (S1, S2, R1 and R2) and Boston (S3, S4, R3 and R4) markets as known commercial brands.

### 2.2. Method

Separations were performed with an Agilent capillary electrophoresis system (Waldbronn, Germany) equipped with a ZETALIF 2000 LIF detector (Picometrics, Montlaur, France). RF was detected with an excitation of 488 nm and emission of 520 nm by an Ar-ion laser. The data processing was carried out with the Agilent ChemStation software. The separation was performed at 25 kV. The temperature was set at 25 °C. Injections were made at 50 mbar for 6 s. The fused-silica capillary used for separation experiments was 50 μm ID and was obtained from Polymicro Technologies (Phoenix, AZ, USA). The total length of the capillary was 67 cm and the length to the detector was 50 cm. The new fused-silica capillary was conditioned prior to use by rinsing with 1 M NaOH for 30 min and with water for 10 min. The capillary was flushed successively by 0.1 M NaOH for 2 min, water for 2 min, and buffer for 5 min, at the beginning of every working day and between runs.

Tea leaves were placed in a pot containing boiling water and the pot was incubated in a water bath for 5 min. Incubation time was optimized by checking the degradation of riboflavin standard in hot

water vs. time, as explained in Results. After filtration of tea leaves from the hot water, tea infusions were directly injected in to the capillary column.

## 2.3. Determination of Total Phenolics (TPs)

The total phenolics of each infusion were determined by using the Folin-Ciocalteu method [18]. For each type of infusion, 300 μL was mixed with 1.5 mL of Folin-Ciocalteu's reagent (1:10 diluted with water) and 1.2 mL of sodium carbonate solution (7.5% *w/v*). The mixture was allowed to stand for 10 min at room temperature until a stable color was obtained. The absorbance of 1/10 fold diluted samples were measured by a Shimadzu UV-1800 spectrophotometer (Shimadzu Scientific Instruments, North America) at 760 nm. Results were expressed as gallic acid equivalents (GAE) in mg/g. The calibration equation for gallic acid was $y = 49.582x - 0.0185$ ($R^2 = 0.995$).

## 2.4. FRAP Assay

The ferric-reducing antioxidant power (FRAP) of the infusions was determined, following the method of Benzie and Strain [19]. The FRAP reagent was prepared containing 1:1:10 ratio of 10 mM 2,4,6-tripyridyl-s-tri-azine (TPTZ) solution in 40 mM HCl, 20 mM $FeCl_3$ and 0.3 M acetate buffer at pH 3.6, and warmed to 37 °C for 10 min. prior to use. The mixture containing 100 μL sample, 100 μL deionized water, and 1.8 mL FRAP reagent was incubated at 37 °C for 10 min. The absorbance of the 1/10 fold diluted mixture was measured by a Shimadzu UV-1800 spectrophotometer at 593 nm. Results were expressed as μmol $Fe^{+2}$/g. The calibration equation for $FeSO_4·7H_2O$ was $y = 20.044x - 0.0373$ ($R^2 = 0.999$).

## 2.5. Statistical Analysis

The statistical analysis was applied to the RF content, TPs, and FRAP values of tea samples. The significant differences between the mean values with $p < 0.05$ were evaluated by a one way analysis (ANOVA) test and the Duncan's new multiple range test using XLSTAT 2017 (Data Analysis and Statistical Solution for Microsoft Excel. Addinsoft, Paris, France (2017)).

## 3. Results and Discussion

### 3.1. Optimization of Separation

Since the pKa value of RF is 9.69, the molecule gains a negative charge in basic solutions and can migrate under an electrical field. Phosphate electrolyte was selected as the separation medium. When the phosphate concentration was changed between 15 and 75 mM, no significant change was observed in peak shapes. On the other hand, while the pH of separation electrolyte increased, the RF peak separated from the electroosmotic peak (negative water peak) and was easily integrated. The fluorescence intensity of RF was maximum at pH 9.9. Finally, 30 mM phosphate at pH 9.9 was selected as the optimal medium for separation and detection.

### 3.2. Optimization of Extraction

RF is slightly soluble in water. One g dissolves in 3–15 L water, depending on the crystal structure (Sigma-Aldrich, Steinheim, Germany). RF is heat stable but very sensitive to light [20]. Considering the very small content of RF in food products, it can be expected that hot water will withdraw the RF in tea leaves. In order to check the stability of RF in hot water vs. time, a RF standard was dissolved in hot water (100 °C). This solution was left in a water bath for different times and riboflavin content of these solutions was determined by the CE-LIF method. Figure 1 shows the comparative results.

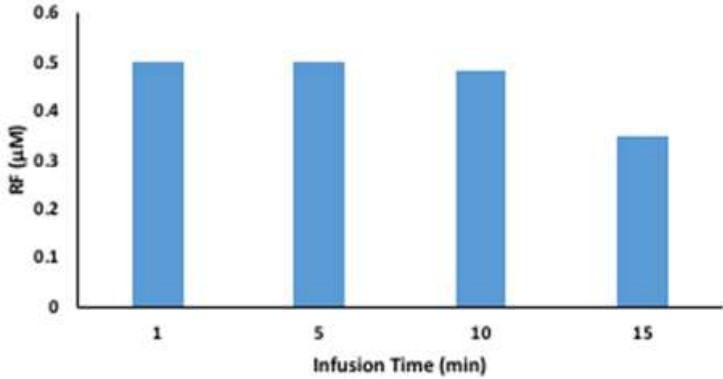

**Figure 1.** Riboflavin amount in hot water vs. infusion time.

As seen from Figure 1, at the end of the 5 min incubation time, we did not observe any decrease in RF concentration in hot water. The decrease of RF content with time after 5 min is probably due to light sensitivity of the molecule. Thereby, 5 min was applied as the optimal incubation time for all tea infusions. 50 mL of boiling water was poured on 1 g of tea leaves and the pot was incubated in a water bath for 5 min. In fact, this is the traditional tea brewing method in homes and coffee houses in Turkey, and 5 min is the accepted time to obtain a good tea infusion.

After filtration of tea leaves from hot water, tea infusions were directly injected in to the capillary column. Figure 2 shows a representative electropherogram of one sage herbal tea infusion (S3). As seen from the electropherogram, the RF peak came in less than 4 min. Since infusions were directly injected without any purification or derivatization step, the analysis method of RF was very short and simple.

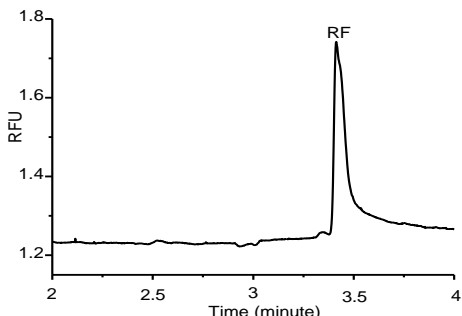

**Figure 2.** Electropherogram of 1/2 diluted sage herbal tea (S3). Conditions: 50 µm × 50 cm capillary, 50 mbar 6s injection, 25 kV running voltage, 30 mM phosphate buffer at pH: 9.9.

*3.3. CE Method Validation*

The calibration curve of RF was linear between 0.01–5 µM concentration ranges. The calibration equation was calculated as y = 0.9544x − 0.0375 ($R^2$ = 0.999). The limit of dedection (LOD) value was obtained from the software of the CE instrument. The calculation was based on the LOD valuebeingthe concentration corresponding to the baseline average noise of electropherogram taken from three different baseline areas. The LOD of the method for RF was found as 1.08 ng/mL. The limit of quantification (LOQ) was given as ten times the average noise as 3.58 ng/mL. For testing the precision of the method, the RF standard solution was injected 5 times in one day. For day-to-day reproducibility, the same solution was injected five times in three different non-consecutive days. In the same day, precision of the corrected peak areas (%RSD; relative standart deviation) was 2.48%. Between days, the precision value was 4.58%.

The recovery experiments were done with one herbal tea sample. The infusion was spiked with standard RF solution atthree different spike levels at the beginning of the extraction process. Satisfactory recovery for RF was obtained as between 99.7 and 106%.

### 3.4. Riboflavin Content of Tea Samples

The RF content of tea infusions are given as the averages of three infusions with their standard deviations in Table 1.

**Table 1.** RF content of tea infusions (**B**: Black, **G**: Green, **S**: Sage, **R**: Rosemary).

| | RF (μg/g ± SD) * | | RF (μg/g ± SD) * |
|---|---|---|---|
| B1 | 3.34 ± 0.19 [d] | S1 | 5.36 ± 0.72 [e] |
| B2 | 0.58 ± 0.16 [abc] | S2 | 5.22 ± 0.29 [e] |
| B3 | 1.57 ± 0.06 [bc] | S3 | 6.18 ± 0.31 [e] |
| B4 | 1.07 ± 0.05 [abc] | S4 | 10.36 ± 0.79 [f] |
| G1 | 3.26 ± 0.46 [d] | R1 | 2.87 ± 0.06 [d] |
| G2 | 2.80 ± 0.10 [d] | R2 | 0.42 ± 0.08 [ab] |
| | | R3 | 1.72 ± 0.35 [c] |
| | | R4 | 0.34 ± 0.01 [a] |

* Means ± standard deviations. Different letters in the same lines are significantly different at the 5% level ($p < 0.05$).

As seen from Table 1, the RF content changes between 0.34 and 10.36 μg/g for all tea samples. Amongst the tested black tea samples, the B1 sample which contains tea leaves fromthe East Black Sea region of Turkey was found to have the richest RF content. The RF content of the blends tested are smaller than in this tea. The RF content of two green tea samples were higher compared to the RF content of black teas. The RF content of rosemary samples were rather similar to the RF content of black tea blends. However, the RF content of all sage teas were substantially higher than those of black and green teas, and rosemary infusions. S4 especially contained significantly higher RF.

### 3.5. Total Phenolics (TPs) and Antioxidant Capacities

The TPs and FRAP values of tea infusions are given in Table 2. TPs ranged from 4.91 to 114 mg GAE/g dry tea leaves for all tea types tested. Green tea samples hadthe highest TPs compared to both black tea and herbal tea samples.

**Table 2.** The TPs and FRAP values of tea infusions. (**B**: Black, **G**: Green, **S**: Sage, **R**: Rosemary).

| | TPs (mg GAE/g ± SD) * | FRAP (μmol Fe$^{+2}$/g ± SD) * |
|---|---|---|
| B1 | 73.55 ± 2.33 [c] | 462.3 ± 0.6 [b] |
| B2 | 94.21 ± 1.59 [e] | 491.8 ± 4.2 [b] |
| B3 | 67.71 ± 1.68 [d] | 449.2 ± 3.7 [b] |
| B4 | 91.39 ± 1.99 [e] | 479.4 ± 10.2 [b] |
| G1 | 113.6 ± 1.9 [f] | 601.4 ± 56.1 [d] |
| G2 | 112.8 ± 1.8 [f] | 551.5 ± 17.5 [c] |
| S1 | 16.86 ± 0.15 [b] | 64.64 ± 8.65 [a] |
| S2 | 17.14 ± 0.10 [b] | 65.08 ± 2.00 [a] |
| S3 | 17.27 ± 0.13 [b] | 78.44 ± 2.22 [a] |
| S4 | 17.97 ± 0.2 [b] | 81.86 ± 0.32 [a] |
| R1 | 17.16 ± 0.62 [b] | 81.34 ± 3.81 [a] |
| R2 | 4.91 ± 0.06 [a] | 63.84 ± 0.84 [a] |
| R3 | 17.14 ± 0.23 [b] | 76.62 ± 1.37 [a] |
| R4 | 5.19 ± 0.02 [a] | 65.89 ± 0.84 [ab] |

* Means ± standard deviations. Different letters in the same lines are significantly different at the 5% level ($p < 0.05$).

The FRAP values of the tested teas ranged between 449–492 μmol/g for black tea samples and 552–601 μmol/g for green tea samples. Benzie and Szeto reported FRAP values for 25 types of teas, ranging between 132–654 μmol/g for black teas and 272–1144 μmol/g for green teas [21]. The FRAP values for black and green tea infusions found in this study are in agreement with these reported FRAP

values. The antioxidant capacities of sage and rosemary teas ranged between 63.8–81.9 µmol/g for 8 herbal tea infusions.

TPs and the antioxidant capacities of black and green tea infusions are obviously higher than those of herbal tea infusions. As expected, there is a strong correlation (0.985) between TPs and the antioxidant capacities of all tea infusions tested in this study. However, there is no correlation between TPs or antioxidant capacity and RF content of teas. Whereas green teas have the highest TPs and antioxidant capacity, the highest RF content was determined to be in sage infusions.

Hu and coworkers reported 2.8 and 5.4 µg/g RF for two green tea samples, which is in agreement with our reported values [14]. RF content of several foods have been reported in the literature. Cataldi et al. have reported the RF content of 8 vegetables ranging between 0.34–1.67 µg/g [12]. The RF content in five milk samples having different animal origins were reported to be between 101 and 175 µg/100 mL, in 2 white wine samples as 12 and 13 µg/100 mL, in raw egg white as 3.8 µg/g, and in raw egg yolk as 3.2 µg/g [13]. The RF in 12 commercial beers was reported as 13–28 µg/100 mL [15]. The RF content of honeys was reported to change in a wide range as from non-detectable to 18.04 µg/g [16]. RF content of five saffron samples from two of the biggest producers in the global market (Iran and Spain) were reported in the range of 5.02–13.86 µg/g [17].

Assuming a tea brew obtained from 2 g of dry tea (around the mass of dry tea in one tea bag) and 200 mL of hot water (the volume of a tea mug), our reported RF values for dry tea samples of 0.34–10.36 µg/g correspond to 0.34–10.36 µg RF/100 mL infusion. When the RF content of tea samples found in this study are compared to the RF content of green vegetables, milk, egg, wine, beer, honey, and saffron samples, known as important RF sources, it can be seen that tea infusions are also important dietary sources of vitamin B2.

## 4. Conclusions

A fast, simple, and sensitive CE-LIF system was used to determine the riboflavin (vitamin B2) content of 14 tea infusions including black, green, sage, and rosemary teas. The RF content of all tea samples suggests that tea infusions are amongst important dietary sources of RF for prevention of diseases caused by vitamin B2 deficiency.

**Author Contributions:** Both of theauthors designed the experiments; F.T. has done experiments and also calculated-analyzed the data; F.T. wrote the manuscript; F.B.E. supervised the research and wrote and edited the manuscript.

**Funding:** This research received funding from the Research Foundation of Istanbul Technical University.

**Conflicts of Interest:** The authors declare no conflict of interest.

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
