# Peer review of "Determination of Vitamin B2 Content in Black, Green, Sage, and Rosemary Tea Infusions by Capillary Electrophoresis with Laser-Induced Fluorescence Detection"

_beverages, doi:10.3390/beverages4040086_

Reviewer 1 Report

In this paper, the Vitamin B2 contents of tea plants was determinated by using the CE-LIF method. It provided useful information. However, some questions need to be solved.

 1.Line 64-65

“Two Sage (S1 and S2) and two Rosemary (R1 and R2) dry herb samples were purchased from Istanbul markets as known commercial brands. Two Sage (S3 and S4) and two Rosemary (R3 and R5) dry herb samples were obtained from Boston-USA markets as known commercial brands.”.

Two sentences were the same besides of the purchase region. Please modified it.

 2.Line 72-73

“RF was detected with an excitation at 488 nm and emission at 520 nm by an Ar-ion laser. The data processing was carried out with the Agilent ChemStation software.”

Please describe the method about the calibration equation or others about the RF determination  

 3.Line 139-141.

“The limit of detection (LOD) was calculated as 3 times of the average noise taken for three different baseline areas and found as 1.08 ng/mL. The limit of quantification (LOQ) was given as ten times the average noise as 3.58 ng/mL”

All LOD calculation has its assumption. Please cite the reference of the LOD calculation equation and explain why to use this equation.

 4.Table 1. RF content of tea infusions and Table 2. The TPs and FRAP values of tea infusions

 The style of these tables need to be revised.

 5.How about the influencing factors of the RF values. The effect of soaping time was explained in Figure 1. Please illustrate other conditions and explain how to use the new technique to detect these factors.

Author Response

Response to Reviewer 2 Comments

Ref.:  Manuscript ID: Beverages-382638

Determination of Vitamin B2 contents in Black, Green, Sage, and Rosemary Tea Infusions by Capillary Electrophoresis with Laser-Induced Fluorescence Detection

Dear Editor,

 I thank the Reviewers comments. I answered all of the comments and questions and also all of the modifications and additions required have been added to the manuscript. I hope that this manuscript will now be accepted for publication.

I thank you for your time and attention.

Sincerely,

Dr. Filiz TEZCAN

Point 1:Line 64-65

“Two Sage (S1 and S2) and two Rosemary (R1 and R2) dry herb samples were purchased from Istanbul markets as known commercial brands. Two Sage (S3 and S4) and two Rosemary (R3 and R5) dry herb samples were obtained from Boston-USA markets as known commercial brands.”.

Two sentences were the same besides of the purchase region. Please modified it.

Response 1 :The sentences were modified as “Sage and Rosemary samples were purchased from Istanbul (S1, S2, R1 and R2) and Boston (S3, S34, R3 and R4) markets as known commercial brands.

Point 2: Line 72-73

“RF was detected with an excitation at 488 nm and emission at 520 nm by an Ar-ion laser. The data processing was carried out with the Agilent ChemStation software.”

Please describe the method about the calibration equation or others about the RF determination  

Response 2:Linear concentration range and equation of calibration curve was already given in the in the CE Method Validation section.

Point 3:Line 139-141.

“The limit of detection (LOD) was calculated as 3 times of the average noise taken for three different baseline areas and found as 1.08 ng/mL. The limit of quantification (LOQ) was given as ten times the average noise as 3.58 ng/mL”

All LOD calculation has its assumption. Please cite the reference of the LOD calculation equation and explain why to use this equation.

Response 3: The LOD value was obtained from the software of CE instrument. The calculation was based on that LOD is the concentration value corresponding to the baseline average noise of electropherogram taken from three different baseline areas. This explanation was added to the text.

Point 4: Table 1. RF content of tea infusions and Table 2. The TPs and FRAP values of tea infusions

 The style of these tables need to be revised.

 Response 4: Table 2 was corrected. The styles of Tables were revised.

Point 5:How about the influencing factors of the RF values. The effect of soaping time was explained in Figure 1. Please illustrate other conditions and explain how to use the new technique to detect these factors.

Response 5: Riboflavin is heat stable but very sensitive to light.  In order to extract riboflavin from tea leaves, we have to use hot water. So we checked standard riboflavin concentration in hot water vs time. As expected, concentration of added riboflavin standard did not change with the temperature of water, however, riboflavin concentration decreased with time probably due to the degradation of the molecule with light.

Reviewer 2 Report

Line 27-28 the beneficial properties linked to tea consumption should be better described and more references should be inserted.

Line 29 Add reference

Line 30 Add reference

Line 31-34 the three types of tea should be described with major details and differences should be marked.

At line 35 the authors should describe the profile of bioactive compounds and antioxidant properties of tea prior to address attention on Riboflavin.

Lines 39-40 a small overview of methodologies for identification and quantification of riboflavin in tea should be added.

In Methods. Were the analysis carried out on tea infusion? the authors should describe the procedure.

Check line 97. The authors should explain the choice of 10 min for incubation in FRAP assay

 The different subparagraphes of Results should be introduced.

The paraghraph 3.2. should be reorganized and written in a clearer manner.

line 114-117 should be rewritten. The procedure for standards should be described also in Methods in addition to infusion procedure.

Line 129-131 this step of procedure should be reported also in Method.

The results in 3.4 and 3.5 are poorly descibed not compared with previous literature data, statistical analysis is missing, a Table is missing!

Table 1 Statistical analysis is missing.

line 177: add references

Table of Total phenols and FRAP is missing

Line 186-194 should be referred to previous subparagraph

Author Response

Response to Reviewer 1 Comments

Ref.:  Manuscript ID: Beverages-382638

Determination of Vitamin B2 contents in Black, Green, Sage, and Rosemary Tea Infusions by Capillary Electrophoresis with Laser-Induced Fluorescence Detection

Dear Editor,

 I thank the Reviewers comments. I answered all of the comments and questions and also all of the modifications and additions required have been added to the manuscript. I hope that this manuscript will now be accepted for publication.

I thank you for your time and attention.

Sincerely,

Dr. Filiz TEZCAN

Point 1:Line 27-28 the beneficial properties linked to tea consumption should be better described and more references should be inserted.

Point 2: Line 29 Add reference

Point 3: Line 30 Add reference

Point 4: Line 31-34 the three types of tea should be described with major details and differences should be marked.

Point 5: At line 35 the authors should describe the profile of bioactive compounds and antioxidant properties of tea prior to address attention on Riboflavin.

Response 1-5:As an answer for the 5 comments above:

The first two paragraphs of the article were rearranged and bioactive components of tea leaves, differences between tea types, health effects of tea consumption have been explained in details and 5 additional new references were added.

Point 6: Lines 39-40 a small overview of methodologies for identification and quantification of riboflavin in tea should be added.

Response 6:As stated in the manuscript, although a significant number of studies have been reported on tea, sage, and rosemary phenolics, almost no information exists concerning the vitamin contents of these plants. Hu and coworkers reported B2 vitamin in two green tea samples [14].To our knowledge, there is no study on the content of Vitamin B2 in sage and rosemary.

 Point 7: In Methods. Were the analysis carried out on tea infusion? the authors should describe the procedure.

Response 7:The analysis were carried out on tea infusions. As explained under the 3.2. Optimization of Extraction: All tea infusions were obtained in hot water with 5 min. incubation time. 50 mL of boiling water was poured on 1 g of tea leaves and the pot was incubated in a water bath for 5 min.

Point 8: Check line 97. The authors should explain the choice of 10 min for incubation in FRAP assay

 Response 8:It was followed the procedure given the related literature (Benzie and Strain [19].)

Point 9: The different subparagraphes of Results should be introduced.

Response 9:Results were given under 5 subtitles.

Point 10: The paraghraph 3.2. should be reorganized and written in a clearer manner.

Response 10:This paragraph was reorganized in order to clarify the selection of optimal infusion time.

Point 11: line 114-117 should be rewritten. The procedure for standards should be described also in Methods in addition to infusion procedure.

Response 11:Lines 114-117 were rewritten. The procedure for standards was mentioned also in Methods.

Point 12: Line 129-131 this step of procedure should be reported also in Method.

Response 12:This step of procedure was reported also in Method

Point 13: The results in 3.4 and 3.5 are poorly descibed not compared with previous literature data, statistical analysis is missing, a Table is missing!

Response 13:We found only one report on the riboflavin contents of tea samples. The comparison of this report has been already given. Table 2 was added. 

Point 14: Table 1 Statistical analysis is missing.

Response 14:Statistical analysis were done and added to the Tables and also the explanation was added to the text in the new section 2.5. Statistical Analysis The statistical analysis was applied for the RF contents, TPs, and FRAP values of tea samples. The significant differences between the mean values at p˂0.05 were evaluated by one way analysis (ANOVA) and the Duncan's new multiple range test using XLSTAT 2017 (Data Analysis and Statistical Solution for Microsoft Excel. Addinsoft, Paris, France (2017)).

Point 15: line 177: add references

Response 15:Line 177 gives the results of Total Phenolics and Antioxidant Capacities of teas in our study. The relating references for these method were already given in Materials and Methods Section.

Point 16: Table of Total phenols and FRAP is missing

Response 16:The TPs and FRAP values of tea infusions was added to the text as Table 2.

Point 17: Line 186-194 should be referred to previous subparagraph

Response 17:We though that these lines are at correct place.

Round  2

Reviewer 2 Report

The authors have improved the manuscript that it is now suitable for publication

Author Response

Ref.:  Manuscript ID: Beverages-382638

Determination of Vitamin B2 contents in Black, Green, Sage, and Rosemary Tea Infusions by Capillary Electrophoresis with Laser-Induced Fluorescence Detection

Dear Editor,

Minor spellchecking and grammar is controlled again. Required revisions are showed by highlighted yellow colour. It’s a pleasure for us to get good news from you about acceptance of our research.

I thank you for your time and attention.

Sincerely,

Dr. Filiz TEZCAN

02 November, 2018